# Emergency department performance assessment using administrative data: A managerial framework

**Anastasiia Soldatenkova, Armando Calabrese, Nathan Levialdi Ghiron, Luigi Tiburzi** *

Dipartimento di Ingegneria dell'Impresa Mario Lucertini, Università degli Studi di Roma "Tor Vergata", Rome, Italy

* luigi.tiburzi@uniroma2.it

**Data Availability Statement:** All relevant data are within the manuscript and its Supporting Information files (S1 Table and S2 Table).

## Abstract

Administrative data play an important role in performance monitoring of healthcare providers. Nonetheless, little attention has been given so far to the emergency department (ED) evaluation. In addition, most of existing research focuses on a single core ED function, such as treatment or triage, thus providing a limited picture of performance. The goal of this study is to harness the value of routinely produced records proposing a framework for multidimensional performance evaluation of EDs able to support internal decision stakeholders in managing operations. Starting with the overview of administrative data, and the definition of the desired framework's characteristics from the perspective of decision stakeholders, a review of the academic literature on ED performance measures and indicators is conducted. A performance measurement framework is designed using 224 ED performance metrics (measures and indicators) satisfying established selection criteria. Real-world feedback on the framework is obtained through expert interviews. Metrics in the proposed ED performance measurement framework are arranged along three dimensions: performance (quality of care, time-efficiency, throughput), analysis unit (physician, disease etc.), and time-period (quarter, year, etc.). The framework has been judged as "clear and intuitive", "useful for planning", able to "reveal inefficiencies in care process" and "transform existing data into decision support information" by the key ED decision stakeholders of a teaching hospital. Administrative data can be a new cornerstone for health care operation management. A framework of ED-specific indicators based on administrative data enables multi-dimensional performance assessment in a timely and cost-effective manner, an essential requirement for nowadays resource-constrained hospitals. Moreover, such a framework can support different stakeholders' decision making as it allows the creation of a customized metrics sets for performance analysis with the desired granularity.

## Introduction

Overcrowding and, consequently, long length of stay (LLOS) are two of the starkest examples of poor performance in emergency departments (EDs). According to major media companies,

**Funding:** The author(s) received no specific funding for this work.

**Competing interests:** The authors have declared that no competing interests exist.

it is not unusual for prospect patients to wait as long as 12 hours for a bed in Italy (visited on May the 30th, 2022) and in the UK (visited on May the 30th, 2022), just to name two of the many examples that can be found online. The roots of this poor performance outcome and their consequences have been investigated in depth. It is a well-known fact that low resource levels, high number of non-urgent visits, and seasonality are among the most common factors originating poor ED performance [1, 2]. Regarding the consequences, it has been established that the overcrowding resulting from poor ED performance reduces the likelihood of hospitalization [3, 4]. It also negatively affects quality of care, as physician forced to work under extreme pressure are more subject to medication errors [5, 6]. As its direst outcome, all else being equal, poor ED performance increases mortality rates [7, 8], thus reducing quality of care particularly for acutely-ill patients [9].

In a recently published article, Yarmohammadian et al. [10] have reviewed the most widely adopted solutions to de-escalate the problem. Such solutions can be broadly classified into two categories, namely resource increase and resource optimization. Obviously, more resources would lead, *ceteris paribus*, to better outcomes. The alternative is to take the most out of the available ones. One solution in this realm, for example, managed to improve ED performance, i.e. to reduce average LOS and overcrowding, by dividing patients into tracks with different priorities based on predefined clinical parameters [11, 12], also known as FAST tracks [13]. Along this line, more dynamic models have been proposed, which combine a patient's condition with LOS level [14]. Traditionally, all these remedies have been focused on the processes or outcomes of core ED functions (e.g. triage, resuscitation, diagnostics, treatment and disposition of patients) and time targets [15, 16].

A less conventional approach to improve ED performance has been lately proposed by Wachtel and Elalouf [1]. They have argued that, in addition to clinical factors, other less obvious elements may affect it. To elicit these additional factors, Wachtel and Elalouf [1] have collected and subsequently analyzed the effect on ED performance of various clinical and non-clinical variables. Finally, they have developed an algorithm, based on the value of the relevant parameters, that hospital managers can utilize, among other things, to reduce average LOS and overcrowding. Their results indicate that along intuitively reasonable clinical factors, such as a patient's blood test and heart rate upon arrival, one non-clinical variable is also highly significant in determining ED performance, namely the patient's number of accompanying escorts.

Wachtel and Elalouf's [1] approach suggests a paradigm change in trying to improve EDs' performance. In this new setting, the focus shifts from the patient's physical condition to a more holistic perspective, namely the status of the organization, the hospital, and its stakeholders [15, 17]. In Wachtel and Elalouf's [1] case, the patient's number of accompanying escorts affect the agility with which the ED reacts. Thus, an alternative way to frame the problem and enrich it with new insights is to examine those organizational factors that can be used to improve the management of patients.

Along the above line of enquiry, this study develops a framework to use administrative health data to analyze patients' flow through EDs. Administrative health data are records of service provision produced routinely by healthcare providers. They have gained popularity in research due to their numerous advantages, among which availability, low cost, large sample size and population coverage [18]. Administrative data allow for investigation of a wide range of aspects, including surgical interventions, treatments, healthcare access, costs of care and variations in resource use [19], but their crucial role in healthcare research is attributable to their use for performance evaluation of health services [20].

Healthcare providers use their administrative data to quantify and compare the performance of selected aspects of care [21]. Evaluation procedures are based on the calculation of measures and indicators, and the results are used for quality control and improvement as well

as in mandatory reporting [22]. Thus, routine data represent an important source for performance monitoring at different levels, such as macro (national, regional, local) or micro (individual provider, clinical area, etc.).

The goal of this study is to harness the value of routinely produced administrative records by developing a framework for multidimensional performance evaluation of EDs that would support internal decision stakeholders in managing operations. This work responds to the need of a practical tool to be used by ED stakeholders for support in decision-making, allowing for prompt problems detection in care quality and care delivery based on already available information. To the best of the authors' knowledge, the only study offering a practical framework for ED performance management has been conducted by Núñez et al. [23]. The authors proposed a set of 75 performance indicators related to processes carried out by EDs that are relevant for monitoring purposes. However, one serious limitation to the Núñez et al. [23] framework's application is that the data is uses are not available, but need to be purposely collected, thus adding an additional burden on an already resource-constrained system. The framework developed in this study, also validated by a series of interviews with a panel of experts, overcomes this limitation as it uses data that are collected on a daily basis by hospitals' Information Systems.

## Methods

In this research, no administrative record related to any patient has been examined, cited, nor referred to. Data in this article consists of theoretical indicators extracted from published scientific papers (properly cited). On the other hand, for the interviews evaluating the framework, i.e., its indicators, doctors and managers provided their verbal consent. Anyway, the Ethical Committee expressed a favorable opinion on the diffusion of the contents in this manuscript.

In detail, the research method is comprised of two broad phases: 1) the design of a multidimensional framework to harness hospital administrative data; 2) an expert session to evaluate the framework. This method loosely follows Gu and Itoh [24], Mantwill, Monestel-Umaña, and Schulz [25], and Stremersch and Van Dyck [26].

In the first phase, the framework is designed bottom up using the logic suggested by Keegan, Eiler, and Jones [27], which includes a definition of the strategic objectives and a decision of what to measure to reach such objectives. In practice, the following steps have been performed:

1. definition of administrative data and overview of the information they contain;

2. definition of the framework's characteristics from the perspective of ED decision stakeholders, and the corresponding key attributes of measures and indicators;

3. review of the academic literature on ED performance measures and indicators;

4. selection of metrics to include in the framework based on the characteristics defined in step 2);

5. organization of the selected indicators into a framework.

In step 4), particular emphasis has been given to the selection of measures that are easy to calculate using information already available in ED administrative data sources. This is to avoid that the managerial framework be yet another burden causing information overload and poor performance [24].

Regarding the second phase, four stakeholders have been identified for the semi-structured interviews, which took place in May 2022. They represent target users of the framework with

**Table 1. Stakeholders' profiling.**

| Role | Perspective | Interactions with data |
|---|---|---|
| General Hospital Manager | purely administrative | Performance monitoring based on the analysis of aggregated results for strategic decision-making |
| Controller | administrative-operational | Day-to-day data analysis, creation of reports and benchmarking with regional standards. |
| ED Clinical Director | clinical-administrative-operational | Clinical information registration coordination and monitoring of the whole ED department for operational decision making |
| COU (Complex Operating Unit) Clinical Director | clinical-administrative | Clinical information registration and monitoring regarding the health services provided by the units related with ED. |

different roles [28]. This role-wise classification reflects diverse professional figures (administrative, clinical, operational) as well as different ways they interact with data in hospital's information systems. The General Hospital Manager, the Controller, the ED Clinical Director and the COU (Complex Operating Unit) Clinical Director have been selected for the interviews. None of the interviewees mentioned anything that could be used to identify specific patients, clinical records, and/or the interviewees themselves. Interviews are therefore fully anonymous. As a consequence, for the correct interpretation of the assessment results, a general profile of each interview participant is presented in Table 1.

The semi-structured interviews have been conducted individually with each stakeholder in Table 1. Interviews started with a brief description of administrative data, the explanation of study aim and methodology, and the specification of the framework. Then, the participants have been asked to evaluate different aspects of the framework. In one case, the evaluation consisted in applying a binary scale (see S1 Table); in another case, a five-point Likert scale ranging from 1 (highly irrelevant) to 5 (highly relevant) has been used (see S2 Table). For the evaluation, the indicators of the original framework have been aggregated into groups based on the indicators' scope and dimension, such as "Quality patient-related metrics: Adverse events". To illustrate each group, some examples of the indicators belonging to the groups were given together with the questionnaire (see S2 Table).

## Framework's design

This section describes the results of the five steps of method's phase 1).

### Administrative data

Administrative data are records on care delivery that are collected for management rather than research objectives [29]. They are generated at the patient discharge from the hospital or hospital-based facility as part of standard coding procedures [30]. Although databases vary in the design and data covered [31], they typically contain basic administrative information about the patient (e.g. age, gender, race, residence code, admission and discharge date), limited clinical data regarding hospital stay (e.g. patient's conditions, procedures received, vital signs, drugs prescribed), and financials (e.g. source of payment, cost of procedures) [32, 33]. Considering their nature, they present some non-trivial ethical challenges, particularly regarding their storage and accessibility [34].

In health services research, two major types of administrative data are used, namely claims data and hospital discharge abstracts [33]. The former represents billing data to the insurer generated by providers for reimbursement purposes; the latter describes hospital-based services for inpatients abstracted from medical records. The corresponding data sources are called

physician billing and hospitalization databases respectively [35]. The main difference is that claims data files are limited to what is billed to the specific payer and provide information across a range of inpatient and outpatient providers as long as the person remains enrolled, while hospital discharge abstract data sets contain records on hospitalizations and services of patients within that hospital only, regardless of payer [33].

This work investigates performance evaluation with administrative data of ED setting. It therefore considers hospital discharge abstract data; in particular, records from the hospital information system regarding services provided to patients in the ED department.

## Defining framework characteristics and key attributes of measures

Performance assessment tools can be designed to provide both qualitative and quantitative outcomes. The most widespread assessment approach is the calculation of performance indicators, which represent "statistical devices" designed to highlight potential problems in care, usually by demonstrating the distance between the calculated value and a benchmark [36]. The development and application of performance assessment tools differ depending on the stakeholders they aim to support [37]. For instance, clinical safety and treatment efficiency indicators may be of higher priority for clinicians, while patient satisfaction could be a main concern of a hospital administrator [38].

The framework developed in this work takes the perspective of ED stakeholders: managers and physicians, who manage operations and are the decision-makers for care delivery organization and planning. Hence, it is designed to provide care delivery performance monitoring and to enable prompt identification of areas for improvement. Furthermore, considering how resource-constrained hospitals are, the framework is conceived so as its implementation and usage do not represent an additional financial burden and do not require extra resource utilization [24, 39].

The selected performance indicators must possess technical and value attributes (Table 2). The technical attributes mean that the indicators must be timely & simple, costless, and available. The value attributes mean the indicators are also valid, general and scalable. Jointly, the two attribute types ensure that the indicators are easy to apply in real-life contexts and provide enough information for an holistic performance evaluation [24, 36, 40].

The use of administrative data sources guarantees that the indicators are costless and available (they are already being collected); the analysis of the academic literature, presented below, ensures validity; the remaining features (scalability, generality, timeliness, and simplicity) are assessed by the authors together with the interviewees.

**Table 2. The key attributes of performance indicators.**

| Attribute type | Attribute | Description |
|---|---|---|
| Technical | Timely & Simple | Calculations are not time-consuming and do not require staff training |
| | Costless | No investment is required to obtain the data; |
| | | Data can be obtained retrospectively (do not require any follow up) |
| | Available | Data are already available and easy to collect and analyze |
| Value | Valid | The indicators are already used or recommended by the research community; |
| | | they relevant to the activity or the outcomes being measured |
| | General | The indicators evaluate the overall ED performance, rather than a specific clinical condition; |
| | Scalable | The indicators allow internal monitoring as well as comparison between EDs |

## Literature review

This section describes the review of the academic literature to identify the most appropriate metrics (indicators and measures) used for ED performance assessment following the criteria in Table 2. The review is based on the widely accepted protocols developed by Tranfield, Denyer, and Smart [41] and Greenhalgh [42].

Scopus and Web of Science (WoS) have been used in combination to identify papers containing ED performance indicators and measures. The following search strings, "performance indicator*", "performance measure*", "emergency department", "ED" have been looked for in the papers' title, abstract and keywords. No time restrictions have been applied. An additional search has been made to capture those metrics using administrative data. The search strings in this case included, "administrative data", "claims", "routine data", and "performance". In both cases, the terms "overall", "global", "general", and "whole" have been added to capture general-purpose indicators, as indicated in Table 2. Only peer-reviewed research papers and reviews written in English have been considered.

The screening process included the following steps (the resulting number of articles is provided in parentheses): citation extraction from the two databases (n = 108), elimination of duplicates (n = 22), title screening (n = 86), abstract screening (n = 80), and full-text analysis (n = 40). Moreover, each article was required to include metrics with the following additional criteria:

- explicitly referred to as measuring performance;

- designed or applied to ED setting;

- evaluate overall performance;

- not being specific to any clinical conditions (diseases or presenting complaints).

Among the 40 selected articles, 24 fulfilled the additional criteria above. Those were selected for full-text analysis. At this point, the references of each of the 24 articles were scanned for relevant material. Three more articles have been included as a result. The final sample consists of 27 articles.

## Emergency department performance framework

A total of 877 performance metrics have been identified in the 27 studies included in the full-text review. Applying the selection criteria defined above, 224 metrics (measures and indicators) have been extracted and organized into the performance measurement framework presented in Fig 1.

The framework is designed as a cube composed of metrics' groups within the three related measurement dimensions: performance, analysis unit, and time-period. The performance dimension is represented by quality of care and time-efficiency [36, 43, 44]. This classification is drawn from the primary function of an emergency department, as highlighted in the report Institute of Medicine [45]: the provision of high-quality care in a timely way. The analysis unit includes the entities in a performance evaluation procedure, namely physician, nurse, disease, triage category etc. as well as the overall facility level. The time-period allows to set timeframes of performance analysis, such as a month, a quarter, or a year. Overall, the analysis unit and time-period help in data grouping and filtering, while the performance dimension reveals the evaluation domain.

Performance metrics identified in the literature have been organized into five groups within the performance dimension. They are represented in the cells on the cube's top layer. Following performance dimension representation, the "temporal" and "quality" metrics have been

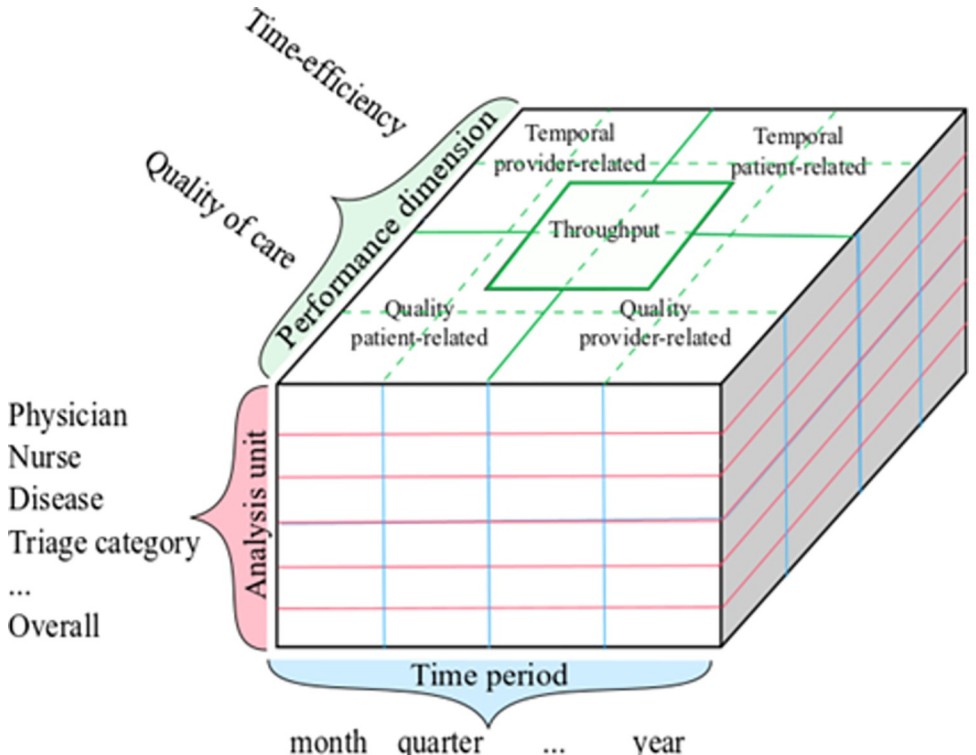

**Fig 1. Schematic illustration of the ED performance measurement framework.**

specified, which in turn have been divided into patient-related and provider-related. This arrangement results in four performance metrics groups: temporal patient-related, temporal provider-related, quality patient-related, and quality provider-related. The quality metrics are those designed for evaluation of the care delivery process and patient outcomes and specified by numbers or proportions of favorable or unfavorable results. The temporal metrics are the waiting times concerning the patient journey and related operations, expressed in time units, such as hours and minutes. The throughput metrics have been placed at the intersection of these four groups. These metrics describe the ED's usage through the patient flow analysis. They do not represent ED performance directly as quality or temporal metrics, but provide information regarding the care process organization. By their nature, they can help reveal the causes of time inefficiencies and bad quality outcomes. In other words, they represent the connection between measurement and management [46].

The cube's underlying idea is that a hospital stakeholder can choose a "slice" at the evaluation and time-period level of interest and calculate different types of performance metrics (e.g., yearly report of guideline adherence by individual physician). The choice depends on the goal of performance measurement and the desired detail. Therefore, the cube represents a flexible instrument that makes it possible to create targeted performance reports. The selected 224 performance metrics, which have been organized into the five performance groups defined above (Fig 1), are presented in Tables 3–7 below. Their description is reported in the original form. Table 3 represents the group of temporal patient-related metrics, where all 41 extracted performance metrics can be referred to as patients' length of stay (LOS) between various points of a care pathway, such as arrival, triage, and care provision. The 23 metrics representing the provider-related time group have been categorized in Decision-related times (12), Interventions and procedural times (8), and Triage time-efficiency (3), as shown in Table 4.

**Table 3. Temporal patient-related metrics.**

| Measure/indicator | Reference | Frequency |
|---|---|---|
| Arrival to Emergency Department (ED) departure/registration to ED departure (admission/discharge/transfer) | [47–49], [50]*3[a], [51–54], [55]*3[a], [56–61] | 19* |
| Triage-treatment in an Emergency Room (ER) bed/ Triage-placement in ED room | [50, 52, 61, 62] | 4 |
| Arrival to physician evaluation | [50, 58] | 2 |
| Arrival to nurse evaluation | [50, 58] | 2 |
| Arrival to treatment/ Door-to-bed time | [50, 55] | 2 |
| Bed to discharge /Treatment to discharge | [50, 52] | 2 |
| Bed-to-provider time | [55] | 1 |
| Arrival-first clinical contact | [47] | 1 |
| Average length of stay (LOS) of ED patients by physician | [50] | 1 |
| Arrival to triage | [50] | 1 |
| Door to important intervention time | [50] | 1 |
| ED room placement to initial md evaluation | [50] | 1 |
| Triage to discharge | [63] | 1 |
| Average triage waiting time ((triage start time—admission end time) / number of patients) | [23] | 1 |
| Triage to physician evaluation time | [50] | 1 |
| Average cycle time of patient per category ((real time of discharge – admission time) / Number of discharged patients) | [23] | 1 |

[a] The frequency is higher than the number of references as some authors discussed the separated metrics that we merged into one

The patient-related quality group is presented in Table 5. It is composed of 42 metrics representing the three types of patient outcomes, namely patient leaves (18), unplanned return visits within different timeframes (18), and adverse events (6). Table 6 shows provider-related quality performance metrics, which represent results of the care delivery and decisions made by medical personnel. It is the largest group with a total of 63 performance metrics, which have been categorized as Treatment Decisions: Procedures (21) and drugs (6), Documentation (21), and Guideline adherence (15). The first category represents decisions regarding performed procedures and drug administration; the second evaluates the patient data collection practice; and the last one helps to reveal the clinical practice behavior of healthcare personnel.

The last group is composed of throughput metrics, which represent counts of patients using the ED system (Table 7). The selected 55 metrics have been categorized into Patient flow by disposition (25): Admission (15), Transfer (8), and Discharge (2), Overall ED occupancy (24); and Patient flow by medical stuff (6).

## Use-case example

The following example helps visualizing how the framework can be used in a real-world scenario. A hospital manager is asked by a director of one of his/her departments the reason as to why numerous patients with a specific clinical condition have recently visited ED and were discharged. Clearly, it is necessary to analyze the performance of the treatment of these patients in the ED. For example, one may want to study how the process was organized to understand if there is room for the implementation of a procedure to schedule a visit for such patients at discharge without affecting ED's overcrowding. In this case, the hospital manager may also want to avoid repetition of some costly exams by sending the diagnostic results

**Table 4. Temporal provider-related metrics.**

| Measure/indicator | Reference | Frequency |
|---|---|---|
| **Provider's decision-related times** | | **12** |
| Decision to admit–admission time | [50, 55] | 2 |
| Arrival to decision to admit | [50] | 1 |
| Arrival to admission order | [56] | 1 |
| Arrival to consultation order | [56] | 1 |
| Consultation order to admission order | [56] | 1 |
| Admission order to ED departure | [56] | 1 |
| Provider-to-disposition time (provider's decision) | [55] | 1 |
| Decision to discharge to discharge time | [50] | 1 |
| Average medical treatment time ((time discharge was decided – time of medical assessment) / Number of discharged patients) | [23] | 1 |
| Decision to transfer to transfer accepted time | [50] | 1 |
| Treatment to decision to discharge/ transfer | [50] | 1 |
| **Intervention and procedural times** | | **8** |
| Time to analgesia/ Time to pain management | [50, 53] | 2 |
| Time from room placement to analgesia | [64] | 1 |
| Time from triage to first analgesia | [64] | 1 |
| Transfer accepted to time left ED | [50] | 1 |
| Timeliness of critical interventions | [50] | 1 |
| Lab turnaround time | [58] | 1 |
| Wait time for labs | [58] | 1 |
| **Triage time-efficiency** | | **3** |
| Begin triage to end triage time | [50] | 1 |
| Average triage time ((triage end time – triage start time) / Number of patients) | [23] | 1 |
| Triage presentation to registration complete time | [50] | 1 |

directly to the clinician with which the follow up visit will be scheduled. However, the hospital does not collect a specific register for this group of patients; on the other hand, it has an administrative database that registers ED attendances. The starting point in the framework application is the study of administrative and clinical information presented in the database. The next step is the choice of performance indicators that can be calculated using the data at hand and are relevant for the investigation.

Fig 2 shows an example of the cube for the described situation: it is sliced at the time-period "single year" and analysis unit "specific diagnosis" dimensions, with the Performance dimension on the front side containing corresponding performance metrics groups and the number of selected indicators in parenthesis. Such high-level representation highlights the dimensions along which performance can be monitored with the available data. At this point, the selected metrics should be calculated and analyzed, enabling the transition from performance measurement to performance management [46].

## Experts' assessment

The framework developed in the previous section has been discussed through a number of semi-structured expert interviews. The main goal of these focus groups has been exploring the thoughts about the framework of key ED figures of a teaching university hospital. All the interviewees are involved in the monitoring of ED performance. They are collectively referred to as decision stakeholders.

**Table 5. Quality patient-related metrics.**

| Measure/indicator | Reference | Frequency |
|---|---|---|
| **Patient leaves** | | **18** |
| Left without being seen (LWBS)/ Left without being evaluated by a physician/ Left before medical assessment | [47, 50, 53, 54, 55, 57, 60, 61, 65] | 9 |
| Left against medical advice (LAMA) | [50, 54, 55, 61] | 4 |
| Left before treatment complete rate (LBTC)/ Left without complete assessment (LWCA) | [50, 59, 61] | 3 |
| Patients left after medical screening exam (LAMSE) rate | [50] | 1 |
| Total abandonment rate after triage (Number of triage patients–Number of discharged patients) / Number of total triage patients | [23] | 1 |
| **Patient returns** | | **18** |
| 72-hour return visits rate | [23, 49, 50, 57, 66] | 5 |
| Unplanned return, 24–72 hours | [50, 61] | 2 |
| Unplanned return, <24 hours, for same or related condition | [50] | 1 |
| Unplanned return within 72 hours for same or related condition | [50] | 1 |
| Unplanned return within 48 hours for same or related condition | [50] | 1 |
| Unplanned return rate to ED or primary care provider | [50] | 1 |
| Return visits resulting in admission to hospital | [50] | 1 |
| Return visit rate, <72 hours with critical diagnosis | [50] | 1 |
| Rate of return visits for patients by physician | [50] | 1 |
| % return to ED within 7 days post ED or observational unit | [50] | 1 |
| ED presentation (return) or direct hospital admission within 28 days (patients discharged) | [67] | 1 |
| Re-attendance rates within 48 hours of discharge | [53] | 1 |
| Readmission with same complaint | [50] | 1 |
| **Adverse events** | | **6** |
| Mortality/ Death in ED | [50, 54, 58] | 3 |
| Mortality rate (within 48 hours of admission to the ED/ and the overall mortality) | [49] | 1 |
| Rate of adverse events | [50] | 1 |
| Rate of deceased patients waiting to be hospitalized | [23] | 1 |

Expert interview is a method of qualitative empirical research to detect expert assessment about a topic that has been considerably used for knowledge production. Compared to multiple experiments, it allows to shorten time-consuming data gathering processes, particularly if the experts possess much practical tacit knowledge [71]. It also serves as an alternative to running *in situ* experiments, which, particularly in the healthcare sector, might result in dire consequences in case of failures.

The results for the questions regarding the awareness of administrative data importance and an overall evaluation of the framework are presented in Fig 3 (S1 Table). The General Hospital Manager is the only stakeholder not fully aware that administrative data can be used for performance assessment. All stakeholders agreed with the proposed clustering of the indicators; they also share the idea that such structure helps evaluating the results. The framework has been judged unanimously as comprehensive for multidimensional performance evaluation and useful to support professional activity, and described as "clear and intuitive", "useful for planning", able to "reveal inefficiencies in care process" and "transform existing data into decision support information".

With respect to ease of implementation in practice, the opinions are different. The Controller and ED Clinical Director indicate that the correct implementation requires effort in terms

**Table 6. Quality provider-related metrics.**

| Measure/indicator | Reference | Frequency |
|---|---|---|
| **Treatment decisions** | | **27** |
| **Procedures** | | **21** |
| Number EKG procedures done | [50, 54, 61] | 3 |
| Number CT scans | [50, 54, 61] | 3 |
| Urinary catheter | [50, 54] | 2 |
| Number Intubations performed | [50, 54] | 2 |
| Number Central line procedures performed | [50, 54] | 2 |
| Number Simple imaging procedures performed | [50] | 1 |
| Number ED ultrasounds performed | [50] | 1 |
| Obtainment of blood cultures before administration of antibiotics | [68] | 1 |
| Number of laboratory studies | [61] | 1 |
| Labs in non-acute patients | [54] | 1 |
| Lumbar puncture counts | [54] | 1 |
| X-ray counts | [54] | 1 |
| Ventilator use rate | [50] | 1 |
| Number of specialty consultations | [61] | 1 |
| **Drugs** | | **6** |
| Inappropriate antibiotic use | [58] | 1 |
| Correct antibiotic use | [50] | 1 |
| % of patients treated with antibiotic in ED | [50] | 1 |
| Administration of antibiotics within 3 hours after ED registration | [68] | 1 |
| Delay to analgesia (% of patients with time from arrival to analgesia ≥60 min) | [64] | 1 |
| Number of drugs prescribed/given per patient at discharge | [50] | 1 |
| **Examination thoroughness and results** | | **21** |
| Accuracy of ED diagnoses | [58] | 2 |
| Examination findings recorded (frequency of records regarding history and physical examination) | [58] | 1 |
| Number or % Of medical records with patient allergy noted | [50] | 1 |
| % Of patients with documented race/ethnicity and language preferences | [50] | 1 |
| Patients evaluated in ED without documentation of vital signs | [50] | 1 |
| Patients with full vital sign documentation | [50] | 1 |
| Incomplete vitals documented | [54] | 1 |
| % Of patients who have documented re-evaluation of analgesia requirements | [64] | 1 |
| Pain score documented | [54] | 1 |
| Specific physiologic parameters for some illnesses | [50] | 1 |
| % Of patients receiving broad spectrum antibiotics | [50] | 1 |
| Most common lab or radiology diagnosis | [50] | 1 |
| Misdiagnosis/missed diagnosis rate | [50] | 1 |
| Inappropriate triage | [58] | 1 |
| Presence of formal triage algorithm | [58] | 1 |
| Prescriptions/patient | [54] | 1 |
| Medications/patient | [54] | 1 |
| Proportion of patients whose pain was assessed | [64] | 1 |
| Proportion of patients receiving analgesia | [64] | 1 |
| % Of patients with administration of any analgesia | [64] | 1 |
| **Guideline adherence** | | **15** |

(*Continued*)

**Table 6.** (Continued)

| Measure/indicator | Reference | Frequency |
|---|---|---|
| Adherence to clinical guidelines | [58] | 1 |
| Compliance with evidence-based guidelines | [50] | 1 |
| Standard compliance rate of treatment times according to triage classification | [23] | 1 |
| % Of patients in severe pain score 7 to 10/10 who receive appropriate analgesia within 20 min of arrival or triage whichever is earlier | [64] | 1 |
| % Of patients in moderate pain score 4 to 6/10 offered analgesia at triage | [64] | 1 |
| % Of patients with severe pain score 9 to 10/10 with no analgesia in the ED | [64] | 1 |
| % Of patients with severe pain score 9 to 10/10 with delay >1 hour in time to analgesia from triage | [64] | 1 |
| % Of patients with severe pain score 9 to 10/10 with delay >1 hour in time to analgesia from placement in room | [64] | 1 |
| % Of patients with delay of ≥1 hour from triage to analgesia | [64] | 1 |
| % Of patients with delay of ≥1 hour from room placement to analgesia | [64] | 1 |
| The percentage of patients seen within the allocated timeframe per triage category | [69] | 1 |
| The proportion of patients admitted or discharged from ED within 4 hour of triage time (National Emergency Access Target [NEAT] criteria) | [63, 70] | 2 |
| The percentage of inpatient admissions leaving ED within 8 hours of ED arrival time | [70] | 1 |
| Percentage of all patients with ED LOS <4 hours if discharged or <8 hours if admitted | [67] | 1 |

of time and bureaucratic procedures' implementation; therefore, they think the framework is not easy to implement. On the contrary, the General Hospital Manager and COU Clinical Director have a more positive attitude about the framework's implementation. During the sessions, only the Controller has replied that no additional collection and digitalization of information on patient visits would be required to measure the performance with the proposed framework.

As a further step, to demonstrate how the framework can be adapted to the needs of the final user(s), participants have been asked to evaluate the framework's content with respect to two dimensions, namely motivation and decision-making (S2 Table). The first one refers to the impact that each indicator has on a personal system of incentives toward professional goals. The last one evaluates the usefulness of information provided by an indicator(s) from a specific group for the management of operations (planning, monitoring, control, etc.) and working context analysis.

The results of the evaluation by each stakeholder are presented in Fig 4. Some indicator groups have been given contradictory evaluations, as in the case of 4.1 (Quality provider-related metrics: Treatment decisions regarding procedures, such as CT scans, X-ray counts, lab studies), which has a low importance for the COU Clinical Director and is of high interest for the General Hospital Manager. This is an expected outcome as each indicator group must be considered within its scope of application as well as its relevance for the goals of each stakeholder. In this case, the indicators in the evaluated group provide information that is relevant from the economic point of view, but is inconsequential for treatment decisions.

Focusing on the pattern of each stakeholder result, it can be noted that for most indicator groups the evaluation in both dimensions coincide, so they are placed on the diagonal. This demonstrates that the framework is relatively balanced as a control system because it impacts equally on the motivation to contribute towards corporate objectives and on decision making activities. However, in the evaluation made by the Controller, almost half of the indicator groups (7 out of 15) have been placed above the diagonal, meaning that the decision-making aspect prevails. This can be explained by the administrative-operational focus of its

**Table 7. Throughput metrics.**

| Measure/indicator | Reference | Frequency |
|---|---|---|
| **Patient flow by disposition:** | | **25** |
| **Admission** | | **15** |
| Admission rate/% Patients admitted | [23, 47, 49], [50]*2[a], [54, 57, 58] | 8 |
| Number of patients admitted from Emergency Department (ED) | [50, 55] | 2 |
| Non-applicable hospitalization rate (Number of C4 and C5 triage patients (non-urgent) that are hospitalized / No of C4 and C5 (non-urgent) triage patients [$C_i$ = category of triage]) | [23] | 1 |
| Intensive Care Unit (ICU) admission rate | [50] | 1 |
| % of critically ill patients admitted | [50] | 1 |
| % of patients admitted with inpatient Length of Stay(LOS) <24 hours | [50] | 1 |
| Patient rate (admitted) per morning/evening/night | [23] | 1 |
| **Transfer** | | **8** |
| Number transfers to and from ED | [50, 54, 55] | 3 |
| Number of patients transferred to another facility after spending >6 hours at initial hospital | [50, 54] | 2 |
| % of critically ill patients transferred | [50] | 1 |
| Transfer rate (Number of transferred patients / Number of total Patients) | [23, 61] | 2 |
| **Discharge** | | **2** |
| Number of patients discharged per hour | [52] | 1 |
| Discharge rate (Number of discharged patients / Number of total Patients) | [23] | 1 |
| **Overall ED occupancy** | | **24** |
| ED census (visits per day, month; by hour of day, at nominated times) | [23], [50]*2[a], [55, 57, 61, 67] | 7* |
| LOS in ED longer than 6 hours | [50, 54] | 2 |
| ED visits per year stratified for main ED, urgent care, fast track, ambulance patient volume, etc. | [50] | 1 |
| Number of patients in waiting room | [47] | 1 |
| Daily boarding hours | [50] | 1 |
| Boarding < 2 hours | [58] | 1 |
| 23h unit usage: patients seen and discharged within 23h, patients admitted but requiring >23 hours, patients admitted to inpatient unit | [50] | 1 |
| Proportion of days with all ED beds occupied | [50] | 1 |
| Visits per bed | [57] | 1 |
| Treatment spaces | [57] | 1 |
| Accessibility of facility after working hours | [58] | 1 |
| % Patients treated | [47] | 1 |
| Proportion of observation patients: discharge vs. admitted | [50] | 1 |
| Number of critically ill patients presenting | [50] | 1 |
| Number of patients sent through ED fast track | [50] | 1 |
| % non-urgents seen in the Urgent Care (UC) or fast track area | [50] | 1 |
| % patients waiting more than 3 hours to see physician | [50] | 1 |
| **Patient flow by medical stuff** | | **6** |
| Admission rate by physician | [50]*2[a], [51] | 3 |
| Patient to doctor per shift | [47] | 1 |
| Patient to nurse per shift | [47] | 1 |
| Percent of patients seen by a physician | [58] | 1 |

[a] The frequency is higher than the number of references as some authors discussed the separated metrics that we merged into one

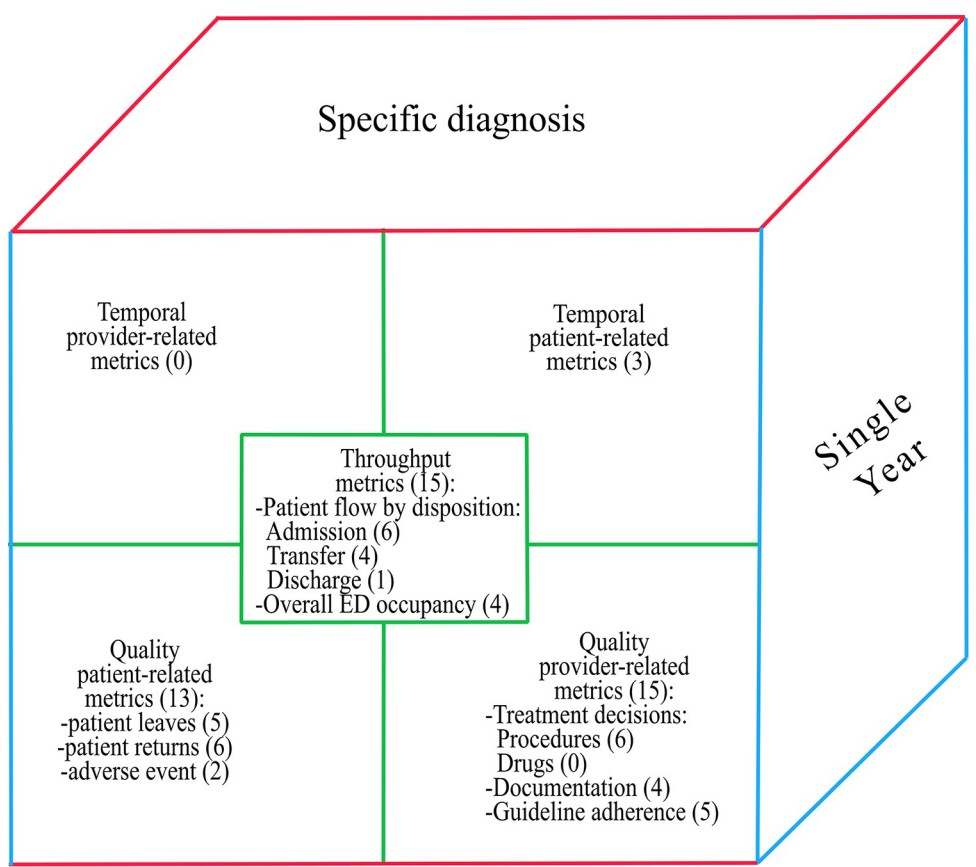

**Fig 2. ED performance measurement framework application: Single year data, patients with specific clinical condition.**

professional activity, which is based on daily data analysis. In addition, such bidimensional evaluation mapping can help in the identification of "core indicators" for each stakeholder. In the present analysis, with a scale from 1 to 5, the most relevant indicator groups are those evaluated 3 or higher in both dimensions (the upper right quadrant in blue in Fig 4).

## Discussion

Little attention has been given so far to the usage of administrative data in emergency departments (EDs). On the one hand, for ED functions, performance measurement is traditionally focused on processes or outcomes (e.g., triage, diagnostics, treatment and disposition of patients) and time targets [15, 16]. On the other hand, as lamented by Madsen et al. [15], performance measurement systems in EDs tend to be narrowly focused on small sets of indicators designed for a specific target (e.g., resuscitation rate). Recently, Wachtel and Elalouf [1] have argued that for EDs too, performance must be measured in a holistic, multi-dimensional way. They have proposed to include also non-clinical variables to monitor the performance of EDs (in Wachtel and Elalouf's [1] case, the number of accompanying escort has been found to be a significant predictor of a patient's LOS).

This work follows Wachtel and Elalouf's [1] approach. In particular, this research designs a framework to harness the value of routinely produced records of administrative data to improve performance management of EDs and health systems in general [1, 20, 72, 73]; 2). This is particularly useful as the adoption of this ED performance measurement framework

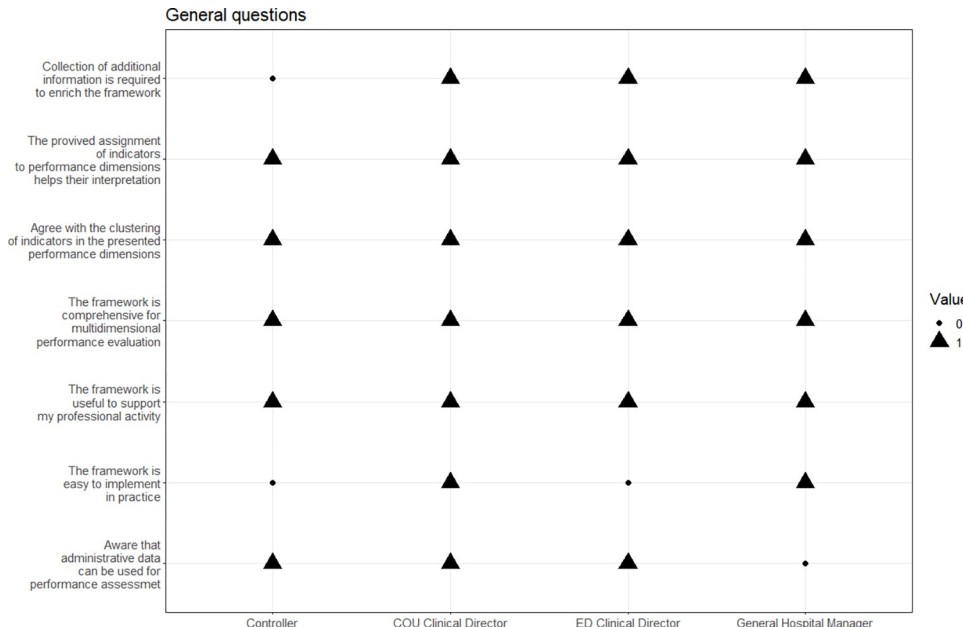

**Fig 3. General framework evaluation.**

will contribute to the use of underexploited, already available health care data [18]. By leveraging routinely registered records, the framework supports tailored operational decision-making at various organisational levels, resulting in supporting ED management. This is relevant for EDs, which are heavily constrained in terms of time and resources [1, 2].

To be used for measuring performance, rather than registering events, administrative data need to be arranged in a way that makes their information relevant. This is one of the framework's major contributions. Clearly, administrative data could be arranged in many different ways. Starting from the classic Donabedian's [74] "structure", "process", and "outcome" model for the assessment of the quality of care, to the multiple dimensions (e.g., appropriateness, expenditure, governance, improved care, clinical focus, efficiency, safety, sustainability and timeliness) identified by, among others, Zaadoud, Chbab, and Chaouch [44], Grimmer et al. [75], and Arah et al. [76], many different frameworks focused on patients' clinical data have been proposed.

The framework in this research abides by the following logic. The core dimension of the framework is evidently performance, as it defines the interpretation of the metrics calculation, while the other two dimensions (analysis unit and time-period) represent their selection bounds. Specifically, performance is measured by taking into account both the analysis unit and the time-period. This three-fold grouping in Fig 1 clusters unorganized administrative data in a way that they can be used for performance measurement because each group produces an unambiguous outcome [36]. In fact, this structure allows for slicing at the desired levels as well as for the creation of metrics sets that allow detailed performance evaluation and reporting (e.g., the monthly throughput for a certain disease). To further improve the framework's ability to provide meaningful results, the selected performance metrics have been organized into multi-level groups, in Tables 3–7, that further narrows down their scope and make them more precise. Finally, the indicators have been selected from the literature. This ensures that each of them is relevant to measure a particular performance aspect [43]. Nonetheless, indicators have been retained only if they were also timely, simple, costless, and available [44].

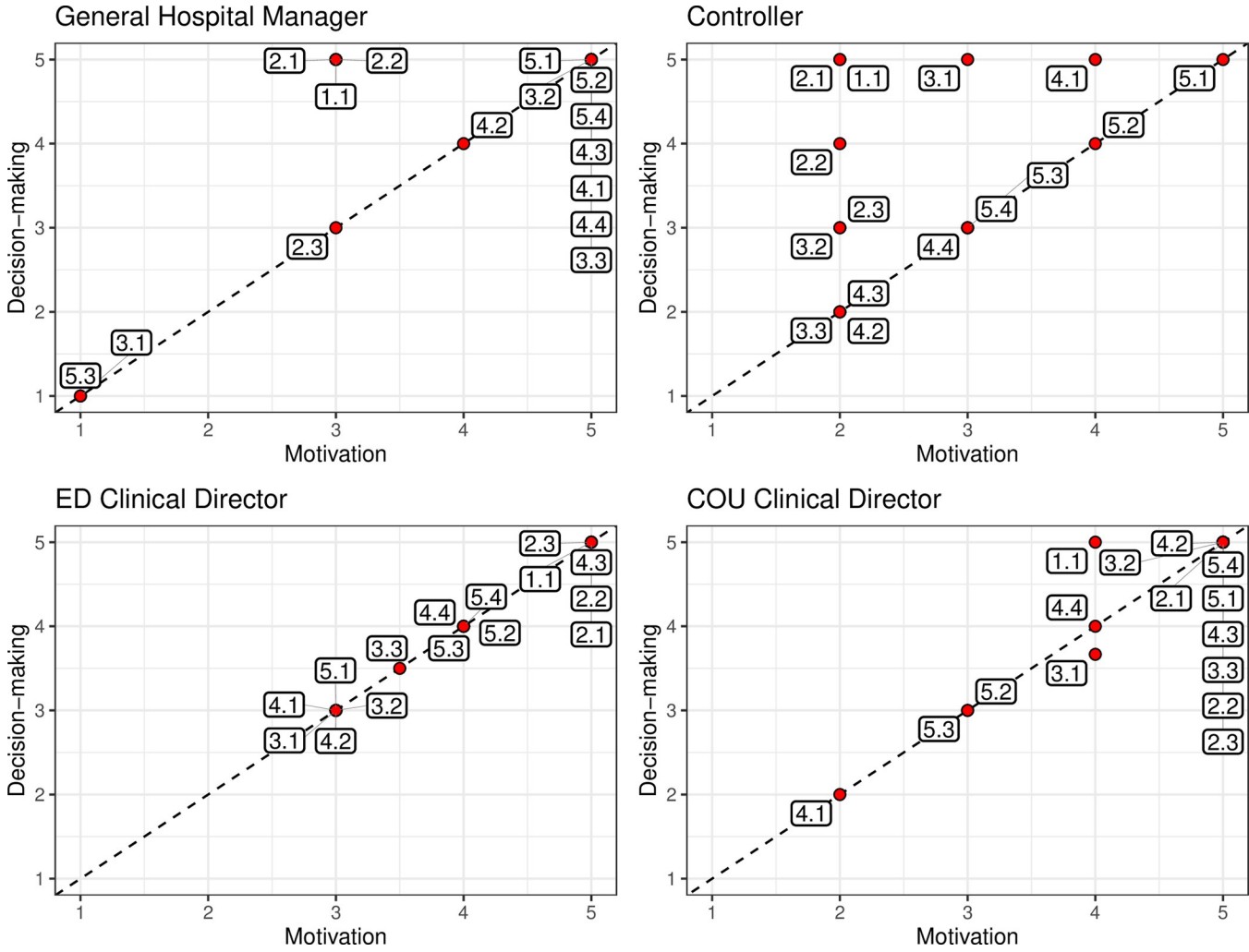

**Fig 4. The framework's content evaluation by stakeholders.**

The rationale is that the framework is conceived to exploit already available data–administrative data–without increasing the burden on a heavily resource-constrained system [24, 39].

Expert interviews have been performed to obtain feedback on the framework. Expert discussion has been paramount to ensure that the framework actually fulfills its role, i.e., it provides on field decision support [28, 43]. During these interviews, the framework has been recognized as a useful tool for performance monitoring and managing operations. Moreover, the interviews showed that the stakeholders' professional role and objectives affect the indicators' perceived importance (Fig 4). This finding is in line with existing literature, which stresses the importance not only of including different indicators, but also of encompassing the perspective of various stakeholders on the same indicators to achieve a comprehensive performance evaluation [77].

Overall, the framework represents a practical tool designed for fast, regular, balanced, and systematic internal ED performance measurement and control, whereby professionals with different roles can select the subsets of indicators mostly tailored to their diverse needs [24, 78]. The framework therefore supports decision-making with respect to specific interests,

responsibilities and objectives at each level of the organization. This contribution timely addresses recognized difficulties of decision-makers to choose a performance measurement system that would suit their needs without increasing their organization's financial burden [79].

## Conclusions and limitations

Administrative data can be a new cornerstone for health care operation management. Using the existing information systems for decision-making support is essential in the resource-constrained hospital environment. This paper proposes a practical framework for performance analysis of hospital emergency departments based on administrative health records. It is aimed to assist decision stakeholders in regular and systematic control of ED performance at the desired level of details in terms of analysis unit, time period and performance dimension. The flexible design allows to identify the core indicators for each target user with respect to his/her professional role and objectives. As a mean of preliminary validation, the framework has been discussed with key ED decision stakeholders of a teaching hospital and it has been judged as comprehensive and useful for managing operations. The current work is a starting point for ED stakeholders to exploit the available administrative data sources to derive valuable performance information for decision-making. In fact, the framework could provide a blueprinting for more advanced, data-driven service design applications, such as BPMN-based approaches for the reengineering of healthcare processes [80, 81].

This research has some limitations. First, in the choice of the metrics to include in the framework only one possible type of administrative data has been considered: the records on care delivery. This implies that the selected indicators refer to care delivery processes and outcomes. At the same time, other types of administrative data sources may be used to evaluate additional performance dimensions, such as structural (attributes of the care settings: equipment, resources etc.) or financial (costs) dimensions [15, 82]. Second, the included metrics have been selected to be computationally simple, as the framework is designed as a practical screening tool to be used by hospital decision makers for performance monitoring and control without any further resource needed. Without this constrain, it would evidently be possible to include other relevant indicators.

## Supporting information

**S1 Table. Experts' general framework evaluation.**
(DOCX)

**S2 Table. Experts' assessment of the framework's content.**
(DOCX)

## Acknowledgments

We would like to express our gratitude to Professor Massimo Federici, Dr. Tiziana Frittelli, Dr. Paolo Furnari, and Professor Jacopo Legramante of the Policlinico Tor Vergata Hospital, who shared their expertise with us during the preparation of this research. We appreciate provided insights and comments regarding the results proposed in the paper. The opinions expressed in this publication are those of the authors and do not necessarily represent those of the Policlinico Tor Vergata Hospital, their officers, or employees.

## Author Contributions

**Conceptualization:** Anastasiia Soldatenkova, Armando Calabrese, Nathan Levialdi Ghiron, Luigi Tiburzi.

**Data curation:** Anastasiia Soldatenkova.

**Methodology:** Anastasiia Soldatenkova, Armando Calabrese, Luigi Tiburzi.

**Resources:** Nathan Levialdi Ghiron.

**Supervision:** Armando Calabrese, Nathan Levialdi Ghiron.

**Writing – original draft:** Anastasiia Soldatenkova, Armando Calabrese.

**Writing – review & editing:** Anastasiia Soldatenkova, Armando Calabrese, Luigi Tiburzi.

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
