## [Decision Letter · Decision Letter 0]

10 Jul 2023

PONE-D-23-12087

Emergency department performance assessment using administrative data: A managerial framework

PLOS ONE

Dear Dr. Tiburzi,

Thank you for submitting your manuscript to PLOS ONE. After careful consideration, we feel that it has merit but does not fully meet PLOS ONE’s publication criteria as it currently stands. Therefore, we invite you to submit a revised version of the manuscript that addresses the points raised during the review process.

We look forward to receiving your revised manuscript.

Kind regards,

Jasna Karacic Zanetti

Academic Editor

PLOS ONE

Journal Requirements:

2. You indicated that ethical approval was not necessary for your study. We understand that the framework for ethical oversight requirements for studies of this type may differ depending on the setting and we would appreciate some further clarification regarding your research. 

Could you please provide further details on why your study is exempt from the need for approval and confirmation from your institutional review board or research ethics committee (e.g., in the form of a letter or email correspondence) that ethics review was not necessary for this study? 

Please include a copy of the correspondence as an "Other" file.

**Additional Editor Comments:**

Please just include ethical challenges regarding administrative data (example

https://www.mdpi.com/2073-431X/12/6/114 )

and send to us back.

Reviewers' comments:

Reviewer's Responses to Questions

**Comments to the Author**

1. Is the manuscript technically sound, and do the data support the conclusions?

Reviewer #1: Yes

Reviewer #2: Yes

2. Has the statistical analysis been performed appropriately and rigorously? 

Reviewer #1: Yes

Reviewer #2: Yes

3. Have the authors made all data underlying the findings in their manuscript fully available?

Reviewer #1: Yes

Reviewer #2: Yes

4. Is the manuscript presented in an intelligible fashion and written in standard English?

Reviewer #1: Yes

Reviewer #2: Yes

5. Review Comments to the Author

Reviewer #1: Many thanks for your good work. I appreciate your efforts in conducting this work. It will help in making the best benefit from generated data in the ED. It's the basis for stakeholder's decisions regarding healthcare delivery and quality assurance.

Reviewer #2: A robust approach should include a performance evaluation of emergency departments using administrative data. Reviewing administrative data can enhance services, clarify patient needs, improve healthcare strategies, increase and keep patients, and develop the patient base. The authors make a point of this.

A teaching hospital's major ED decision stakeholders can use the administrative data collected for planning to identify inefficiencies in the care process and translate existing data into decision support information. It improves emergency room quality assurance even more.

The authors have done well by shining light on this subject matter. Usually, this sort of data may be readily available.

6. PLOS authors have the option to publish the peer review history of their article (what does this mean?). If published, this will include your full peer review and any attached files.

Reviewer #1: No

Reviewer #2: No

---

## [Author Response · Author response to Decision Letter 0]

2 Aug 2023

Journal requirements

We have changed the style of the manuscript according to the guidelines provided in the links above.

2. You indicated that ethical approval was not necessary for your study. We understand that the framework for ethical oversight requirements for studies of this type may differ depending on the setting and we would appreciate some further clarification regarding your research. 

Could you please provide further details on why your study is exempt from the need for approval and confirmation from your institutional review board or research ethics committee (e.g., in the form of a letter or email correspondence) that ethics review was not necessary for this study? 

Please include a copy of the correspondence as an "Other" file.

The study is theoretical, not practical. Actually, no administrative record related to any patient has been seen, nor cited, nor referred to in any way. However, a statement from our Research Ethics Committee has been attached. In the statement, the Committee has specified that the study poses no ethical concerns. Also a copy of their email has been added.

The following statement has been added at the beginning of the Methods:

In this research, no administrative record related to any patient has been examined, cited, nor referred to. Data in this article consists of theoretical indicators extracted from published scientific papers (properly cited). On the other hand, for the interviews evaluating the framework, i.e., its indicators, doctors and managers provided their verbal consent. Anyway, the Ethical Committee expressed a favourable opinion on the diffusion of the contents in this manuscript.

…

The General Hospital Manager, the Controller, the ED Clinical Director and the COU (Complex Operating Unit) Clinical Director have been selected for the interviews. None of the interviewees mentioned anything that could be used to identify specific patients, clinical records, and/or the interviewees themselves. Interviews are therefore fully anonymous.

We have uploaded the supplementary materials (S1 and S2) and pointed to them in the text. The minimal data set consists of S1 and S2.

We have included the approval statement from our Ethical Committee in the methods. As specified above, we have received verbal consent from the interviewees, witnessed by three of the authors. The approval statement from the Ethical Committee has also been uploaded as a file ‘Other’.

We have included captions to the supplementary materials.

Editor

Please just include ethical challenges regarding administrative data (example https://www.mdpi.com/2073-431X/12/6/114 ) and send to us back.

We have included a citation to this work, which we find relevant in our field. Thanks for suggesting it.

---

## [Decision Letter · Decision Letter 1]

12 Oct 2023

Emergency department performance assessment using administrative data: A managerial framework

PONE-D-23-12087R1

Dear Dr. Tiburzi,

We’re pleased to inform you that your manuscript has been judged scientifically suitable for publication and will be formally accepted for publication once it meets all outstanding technical requirements.

Kind regards,

Gokhan Agac, Ph.D.

Guest Editor

PLOS ONE

Additional Editor Comments (optional):

Reviewers' comments:

Reviewer's Responses to Questions

**Comments to the Author**

1. If the authors have adequately addressed your comments raised in a previous round of review and you feel that this manuscript is now acceptable for publication, you may indicate that here to bypass the “Comments to the Author” section, enter your conflict of interest statement in the “Confidential to Editor” section, and submit your "Accept" recommendation.

Reviewer #2: All comments have been addressed

2. Is the manuscript technically sound, and do the data support the conclusions?

Reviewer #2: Yes

3. Has the statistical analysis been performed appropriately and rigorously? 

Reviewer #2: Yes

4. Have the authors made all data underlying the findings in their manuscript fully available?

Reviewer #2: Yes

5. Is the manuscript presented in an intelligible fashion and written in standard English?

Reviewer #2: Yes

6. Review Comments to the Author

Reviewer #2: I commend the author for addressing the concerns raised diligently and thoroughly. The authors attention to detail and commitment to improving the manuscript are evident in the revised version.

I have found no issues Regarding dual publication, research ethics, and publication ethics. The research adheres to established ethical guidelines, and there are no indications of dual publication or research misconduct.

Overall, I am now highly confident in the quality and rigor of your work. I recommend this manuscript for publication without hesitation.

7. PLOS authors have the option to publish the peer review history of their article (what does this mean?). If published, this will include your full peer review and any attached files.

Reviewer #2: No

---

## [Editor Report · Acceptance letter]

18 Oct 2023

PONE-D-23-12087R1 

Emergency department performance assessment using administrative data: A managerial framework 

Dear Dr. Tiburzi:

I'm pleased to inform you that your manuscript has been deemed suitable for publication in PLOS ONE. Congratulations! Your manuscript is now with our production department. 

Kind regards, 

on behalf of

Professor Gokhan Agac 

Guest Editor

PLOS ONE